# An ancestral fold reveals the evolutionary link between RNA polymerase and ribosomal proteins

Sota Yagi [1,2] & Shunsuke Tagami [1,3,4]

Numerous molecular machines are required to drive the central dogma of molecular biology. However, the means by which these numerous proteins emerged in the early evolutionary stage of life remains enigmatic. Many of them possess small β-barrel folds with different topologies, represented by double-psi β-barrels (DPBBs) conserved in DNA and RNA polymerases, and similar but topologically distinct six-stranded β-barrel RIFT or five-stranded β-barrel folds such as OB and SH3 in ribosomal proteins. Here, we discover that the previously reconstructed ancient DPBB sequence could also adopt a β-barrel fold named Double-Zeta β-barrel (DZBB), as a metamorphic protein. The DZBB fold is not found in any modern protein, although its structure shares similarities with RIFT and OB. Indeed, DZBB could be transformed into them through simple engineering experiments. Furthermore, the OB designs could be further converted into SH3 by circular-permutation as previously predicted. These results indicate that these β-barrels diversified quickly from a common ancestor at the beginning of the central dogma evolution.

The central dogma of molecular biology is governed by numerous molecular machines, including DNA polymerases, RNA polymerases, and ribosomes. Despite the detailed understanding of their regulation mechanisms, the evolutionary origins of such complex molecular machines remain obscure.

The evolutions of some pivotal proteins in the central dogma may have originated from the well conserved small β-barrels within their core regions[1]. For example, the core domains of DNA polymerase D (PolD) from euryarchaea and all cellular RNA polymerases are composed of two homologous β-barrels with six strands, "double-psi β-barrels (DPBBs)"[2–4] (Supplementary Fig. 1A). The ribosomal protein L3 (rL3) and several translation factors have a similar but topologically distinct six-stranded β-barrel, "RIFT"[5,6] (Supplementary Fig. 1B). The structures of DPBB and RIFT have two-fold pseudo symmetry, indicating they originated as shorter homo-dimeric peptides[5,7]. Five-stranded β-barrel folds such as "OB" and "SH3" are often found in other ribosomal proteins and translation factors[8–10] (Supplementary Fig. 1C

and D). Given that these β-barrel domains are highly conserved across all extant organisms and play critical roles in replication, transcription, and translation, it is hypothesized that they were among the earliest components of the primordial central dogma machinery[1,2,11].

These four β-barrels (DPBB, RIFT, OB, and SH3) are classified into different folds in the SCOP, CATH, and ECOD protein databases[12–15], as they have distinct topologies. Even so, the partial structure and sequence similarities between these folds have been detected[16]. Over the last two decades, meticulous comparative analysis of sequence motifs and partial structures have independently suggested that the DPBB-RIFT, RIFT-OB, and OB-SH3 pairs diverged from a common ancestral protein[5,6,9,17] (Supplementary Fig. 1). Despite these efforts, no experimental evidence has been provided to demonstrate that such drastic fold transitions could occur via a feasible pathway, probably because of the huge sequence/structure diversity between modern proteins with the different folds, especially between the pseudo-dimeric ones (DPBB and RIFT) and the monomeric ones (OB and SH3).

[1]RIKEN Center for Biosystems Dynamics Research, 1-7-22 Suehiro-cho, Tsurumi-ku, Yokohama, Kanagawa 230-0045, Japan. [2]Faculty of Human Sciences, Waseda University, 2-579-15, Mikajima, Tokorozawa, Saitama 359-1192, Japan. [3]Graduate School of Medicine, Science and Technology, Shinshu University, 3-1-1 Asahi, Matsumoto City, Nagano 390-8621, Japan. [4]International Institute for Sustainability with Knotted Chiral Meta Matter (WPI-SKCM²), Hiroshima University, 1-3-1 Kagamiyama, Higashi-Hiroshima, Hiroshima 739-8526, Japan. ✉e-mail: sota.yagi@aoni.waseda.jp; shunsuke.tagami@riken.jp

Therefore, an experimental reconstruction of the ancient evolutionary process between these β-barrels has been awaited to reveal the profound protein fold evolution before the establishment of the central dogma.

Here, we study fold transition between distinct β-barrels by protein engineering and structural biology technique. We previously reconstructed the evolutionary pathway of the DPBB fold, initiated through the homo-dimerization of a half-sized peptide with about 40 amino acids, followed by gene duplication and fusion[7]. Furthermore, by reducing the amino acid repertoire of the peptide, we have created the homo-dimeric DPBB fold comprising only seven amino acid types (Ala, Gly, Asp, Glu, Val, Lys, and Arg; design-1)(Fig. 1A, B), which could have been synthesized by immature translation systems in early life[7,18]. In this study, by using this simplified DPBB peptide as the starting template, we experimentally reconstructed the evolutionary pathways between the various ancient β-barrel folds in the central-dogma machinery through an unexpected missing link.

## Results

### Conversion of homo-dimeric β-barrels through a missing-link fold

The most simplified DPBB protein we designed previously, design-1 (mk2h_ΔMILPYS), did not fold in the typical buffer conditions (50 mM phosphate, 150 mM NaCl), but crystallized under two different conditions, containing malonate or malic acid ions, and adopted the DPBB

fold in the crystals (Fig. 1A, B)[7]. Here, we report its third type of crystal, formed under different conditions (100 mM Tris, pH8.5, 20% PEG-400, 200 mM lithium sulfate). Interestingly, we could not solve its structure by molecular replacement using the DPBB fold as a model, implying that design-1 had adopted a different conformation in the third type of crystal.

At first, we expected that it adopted a RIFT-like structure because DPBB and RIFT supposedly evolved from a common ancestral homo-dimeric peptide[7]. Indeed, they commonly have (i) six-stranded β-barrel structures, (ii) an internal pseudo-two-fold symmetry, (iii) and a sequence motif "GD-box", although the 1st loop configuration and the β2 direction are different (Supplementary Fig. 2). Following this assumption, we tried to stabilize the unsolved conformation of design-1 by introducing amino acid residues conserved in Phs018, a RIFT protein. Phs018 has high two-fold symmetry and likely retains the properties of the ancient RIFT-fold proteins[5]. Phs018 has remarkable sequence similarity to design-1 (26–35% identity; Supplementary Fig. 2), and we replaced five residues in design-1 with the ones at the corresponding positions of Phs018 (Fig. 1A). AlphaFold2 (AF2)[19] predicted that the resultant mutant, design-2 (Ph1), would fold into a RIFT-like structure, albeit with a low lDDT region in the 1st β-turn (Supplementary Fig. 3). Design-2 was expressed, purified, and analyzed physiochemically. Circular dichroism (CD) and size exclusion chromatography (SEC) experiments indicated that design-2 was folded (not random coil) and had

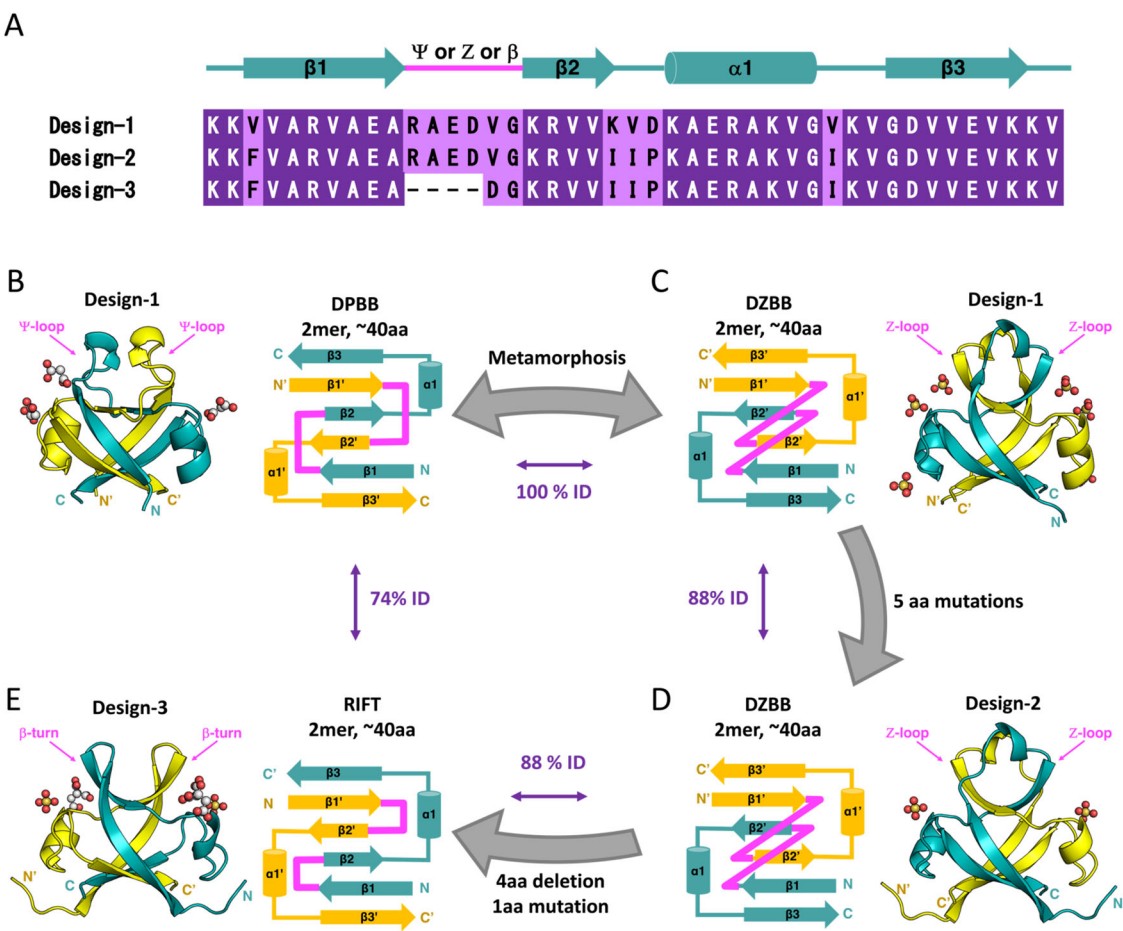

**Fig. 1 | Experimental reconstruction of the evolutionary pathway from homo-dimeric DPBB to RIFT through another barrel fold DZBB. A** The amino acid sequences of the representative designs are aligned with the secondary structure elements. **B–E** The crystal structures and topological schemes of engineered proteins are shown. Single chains in each homo-dimeric structure are colored cyan and yellow. The 1st loops connecting β1-β2 are highlighted in pink in all panels. The sequence identities between the proteins are also shown between their panels. **B** and **C** Design-1 adopted in two folded states, DPBB and DZBB. **D** The five-point mutations stabilized the DZBB fold in design-2. **E** The remodeling of the loop connecting β1-β2 in design-2 resulted in the RIFT-fold protein, design-3.

moderate stability ($T_m$ = 58 °C) (Supplementary Fig. 4). Design-2 also formed crystals under similar buffer conditions to the third type of design-1 crystal, including lithium sulfate.

By molecular replacement with the predicted design-2 model (RIFT fold) and subsequent manual remodeling, the crystal structures of design-1 (third type of crystal) and design-2 were determined (Supplementary note 1). Surprisingly, they both adopted into a unique β-barrel fold that topologically differs from DPBB and RIFT (Fig. 1C, D). Compared to the homo-dimeric DPBB fold, the directions of the β2- and β2'-strands were inverted, resulting in all anti-parallel strand patterns like those in RIFT. However, the 1st loop connecting β1– β2 was rolled up, unlike the simple β-turn in RIFT (Fig. 1C, D, and Supplementary Fig. 5A). As this loop configuration resembles the letter "Zeta" in the topological scheme, we named this β-barrel fold the Double-Zeta β-barrel (DZBB). Therefore, design-1 can fold into two different structures with an identical sequence. The five point-mutations derived from Phs018 (RIFT) in design-2 stabilized the DZBB fold.

To further convert DZBB into RIFT, the sequence forming the Z-loop in design-2 was replaced with two residues (DG, GG, or GD) facilitating β-turn formation[20] (design-3, -4, and -5) (Supplementary Fig. 6 and Supplementary Table 1). CD experiments demonstrated that design-3 and design-5 were partially folded (Supplementary Fig. 7), and design-3 formed well-diffracting crystals. In contrast, design-4 was unfolded but still formed crystals in the presence of sulfate ions. The crystallographic analysis revealed that design-3 and design-4 adopted the homo-dimeric RIFT-fold (Fig. 1E and Supplementary Fig. 5B). Thus, the short In/Del at the 1st loop position is the determinant for the fold transition between DZBB and RIFT (Fig. 1A). The successful experimental conversion from DPBB to RIFT through the DZBB fold, by just a few mutations, indicates that DZBB is a missing link between the ancient homo-dimeric β-barrels in transcriptional and translational proteins (Supplementary Fig. 1).

## Dual-folding of "design-1" induced by small ligands

Metamorphic proteins are a rare protein class that reversibly convert between distinct folded conformations in native conditions[21–23]. Design-1 exhibited the metamorphic property, adapting two different folds, DPBB and DZBB (Fig. 1B, C). Design-1 was originally obtained by substituting three tyrosine residues in the stable parent DPBB protein, design-0 (mk2h_ΔMILPS), composed of eight amino acid types[7]. Thus, these three mutations destabilized the DPBB fold and then allowed it to fold into the DZBB fold, resulting in the dual-folding property. Furthermore, the addition of five-point mutations in design-2 stabilized the DZBB fold and abolished the capability to adopt the DPBB fold. Such metamorphic states like those of design-1 could have existed to bridge between different folds smoothly during drastic fold transitions[22,24].

Interconversion between the two folds also resulted in the distinct domain-swapping states (DPBB: β3β1'β2β2'β1β3'; DZBB: β3β1β2'β2β1'β3'). While domain-swapping with multiple oligomeric states have been observed in some natural and designed proteins[25–27], the rearrangement of the β-strands orientation and drastic changes of intra- and inter-chain interactions in the homo-dimeric structure like design-1 is unusual, extending our knowledge of domain-swapping proteins.

Different ligand molecules probably induced the metamorphism in design-1. In the crystal structure of the DPBB-fold, two malonate ions bind to a positively charged pocket around the α1 helix, and are coordinated by Lys24, Arg27, and Lys33 (Fig. 2A). Alternatively, in the crystal structure of design-1 with the DZBB fold, two additional residues, Arg21, and Arg18' from the other chain, coordinate the two sulfates (Fig. 2B and Supplementary Fig. 8). The sulfate ions may attract these two additional residues in the folding process, and then stabilize the inverted β2-strand in the DZBB-fold.

To test whether these small molecules can facilitate the folding of design-1 in solution, we analyzed its conformational change by

monitoring the binding of the fluorescence probe 8-anilino-1-naphthalenesulfonic acid (ANS). ANS typically binds to a hydrophobic patches of the folding intermediates and molten globules, which changes its fluorescence spectrum. A low concentration of malonates (50–500 mM) did not alter the ANS fluorescence spectra (Fig. 2C). However, the fluorescence signal was increased at over 1,000 mM of malonates, and its peak was blue-shifted (Fig. 2C), implying that high concentrations of malonate induce at least partial folding of design-1. We also found that high concentrations of sulfate ion (≥1500 mM) increased ANS fluorescence (Fig. 2D). Its spectral pattern was slightly different compared to malonates, perhaps due to the difference in the DPBB- and DZBB-folds. The changes in the CD spectrum patterns depending on the sulfate ions were also observed (Supplementary Fig. 9A). These experiments demonstrated that the small ions mediate the folding of design-1, and their types may lead to two different structures (Fig. 2E).

Interestingly, ANS experiments showed that similar ions, phosphate, malic acid, and citrate, also promoted conformational changes (Supplementary note 2 and Supplementary Fig. 10). As these anionic ions probably existed in the primordial cells and on the early Earth[28–31], they might have served as chemical chaperones to enhance the folding of ancient proteins and could have compensated for the low stabilities of evolutionary intermediates during the folding transition (Supplementary note 3 and Supplementary Fig. 7–10).

## The transformation from DZBB to the monomeric OB-fold

In the homo-dimeric DPBB and RIFT structures, the β-strands from a monomer are mostly interlaced with the β-strands from the other chain, and thus these peptides can only fold as homo-dimers (Fig. 1B, E). In contrast, in the structure of DZBB, the secondary structure elements from a single subunit are mostly clustered together as in a monomeric protein, except for the swapped β2 strands (Fig. 1C, D). Surprisingly, a structural similarity search using the DALI software[32] detected high correspondence between the monomeric part of DZBB and OB-fold proteins. In the superimposition of the DZBB and OB proteins, the β1, β2, and β3 strands of DZBB are well aligned with the β1, β3, and β4 strands of the OB-fold protein, respectively (Fig. 3A). In addition, their sequences are partially similar (Supplementary Fig. 11A). The only significant differences between their structures are the presence and absence of a few secondary structural elements (Fig. 3B). The DZBB monomer lacks two β-strands in the OB-fold (β2 and β5), and while the β2 strand is conserved within OB-fold proteins, the β5 strand is poorly conserved or even absent in some OB proteins; e.g., ribosomal protein L2. Helix α1 of the OB-fold (corresponding to α1 of DZBB) is also missing in some OB proteins (e.g., rL2, S17, and S28). Thus, the original OB-fold has been considered to be a four-stranded β-barrel, of which only β2 is absent in DZBB[10].

To demonstrate the hypothesized interconversion between the DZBB-fold and the OB-fold, we created their chimeric proteins by combining the design-1 and OB proteins. The sequences of the OB domains in the rL2 proteins from thermophilic archaea, *Thermococcus kodakarensis* and *Methanopyrus kandleri*, resemble that of design-1, while helix α1 is absent in the rL2 proteins. In particular, the OB-domain of rL2 from *M. kandleri* exhibited a high similarity with design-1 (identity 25%, Supplementary Fig. 11A). Given that design-1 was originally constructed from the DPBB protein from *M. kandleri*[7], the genome of this archaeon might still preserve the evolutionary information of ancient proteins. The sequence region surrounding β2-β3 of the OB-domain in rL2 was incorporated into the corresponding position of design-1, based on the superimposed structures (Fig. 3A). Through this process, we constructed six unique variations by modifying the positions and extents of insertion (design-6–11; Supplementary Fig. 11B and Supplementary Table 1). Biochemical and crystallographic analyses revealed that two of the six chimera proteins, design-6 (tkoL2_v1) and design-9 (mkaL2_v1) (Fig. 3C), adopt four-

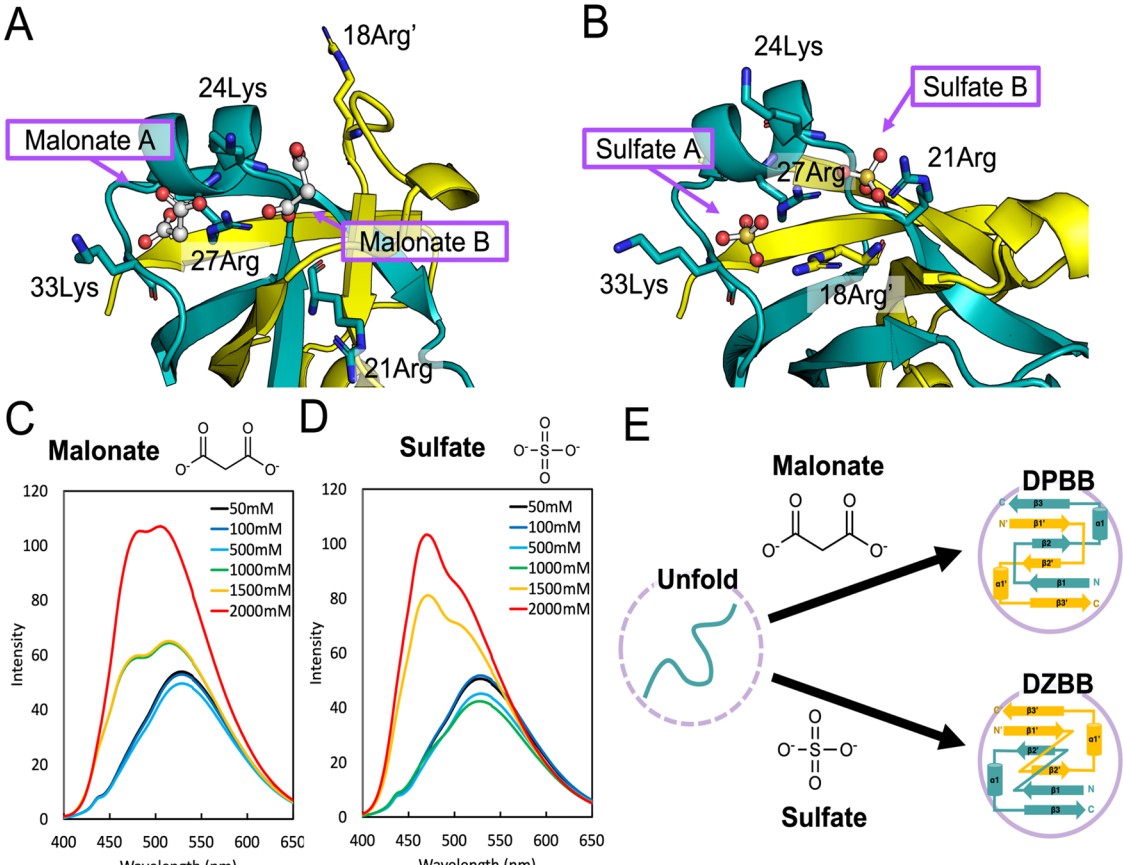

**Fig. 2 | Induction of fold changes by malonate and sulfate ions. A** Two malonates in the crystal structure of chemically synthesized design-1 in the DPBB form. **B** Two sulfates in the crystal structure of chemically synthesized design-1 in the DZBB form. The five residues related to the coordination of ions are shown as stick models. **C, D** The ANS fluorescence spectra of design-1 in the presence of increasing concentrations of (**C**) malonate or (**D**) ammonium sulfate. Similar experiments were performed twice, and representative data are shown. Source data are provided as a Source Data file. **E** The scheme of the dual-folding mechanism of design-1 dependent on the types of ions.

stranded OB-folds (Fig. 3D, Supplementary Fig. 5C and 12). The DZBB and OB fragments comprise approximately 40 and 60% of these chimeric proteins, respectively. Therefore, the monomeric OB-fold could be reconstructed by simply combining the DZBB and OB-fold protein segments, without optimizing the structural interfaces between both parts.

To examine what determines the DZBB−OB transition, additional intermediates have been engineered by sequential mutagenesis and experimental validation steps (design-12–17; Supplementary note 4, Supplementary Fig. 5D, 13A, and Supplementary Table 1). The 2nd generation mutants, design-13 (tkoL2_v1.2) and design-16 (mkaL2_v1.2), have ~60% sequence identities with design-2 (DZBB-fold), and retained moderate thermostability (Fig. 3C and Supplementary Fig. 14). In these designs, only a short segment was from rL2 (segment 1: from the middle of β1 to the start of β3), and the other parts are from design-2 (Fig. 3C and Supplementary Fig. 13A). Helix α1 (segment 2) was also omitted. The crystallographic analysis revealed that design-13 still adopts the four-stranded OB fold (Fig. 3E), indicating that segment 1, but not segment 2, could serve as the determinant for the fold transition between the DZBB and OB structures. To test this assumption, we conducted the reverse engineering by replacing segment 1 of design-13 and design-16 with the 8-amino acid sequence forming the Z-loop of design-2 (the 3rd generation mutants: design-18 (tkoL2_v1.2_Z) and design-19 (mkaL2_v1.2_Z))(Fig. 3C, Supplementary Fig. 13B, Supplementary Table 1). SEC and CD experiments demonstrated that both mutants folded and had moderate thermal stabilities (Supplementary Fig. 15). The crystal structures of design-18 and design-

19 revealed that they adopt the DZBB fold even without the α1 region (segment 2) (Fig. 3F and Supplementary Fig. 5E). These results demonstrated that the segment 1 is sufficient to archive the fold-change from DZBB to OB.

Furthermore, we verified that the 13 a.a. sequence forming the flexible β-turn in segment 1 in the OB-fold designs could be shortened to 7 a.a. (Supplementary note 4, Supplementary Figs. 16–18, and Supplementary Table S1). Taken together, very short In/Del and a few point mutations are the determinants in the transition between the DZBB and OB folds. This facile interchangeability between DZBB and OB folds suggests that such a drastic fold transition likely occurred in the early evolutionary stage of life.

**Transformation from OB to SH3**

Given the high similarity between the OB and SH3 folds, they are presumed to have evolved from a common ancestral protein (Supplementary Fig. 1)[9,10]. Loren Williams' group suggested that the four-stranded core fold of OB could have transformed to SH3 by a simple circular permutation (or vice versa)[10]. Following this evolutionary hypothesis, we tried to convert the reconstructed OB proteins to the SH3 fold. The fourth strand of design-6 and design-9 was trimmed and connected to the N-terminal end by two residues "DG," an ideal sequence to form a β-turn[20] (design-23: tkoL2_v1_SH3; design-24: mkaL2_v1_SH3)(Supplementary Table 1). While design-23 had a random-coil structure, design-24 exhibited a CD spectrum for a folded protein and remained almost intact even at 90 °C (Supplementary Fig. 19). We also determined its crystal structure and confirmed that

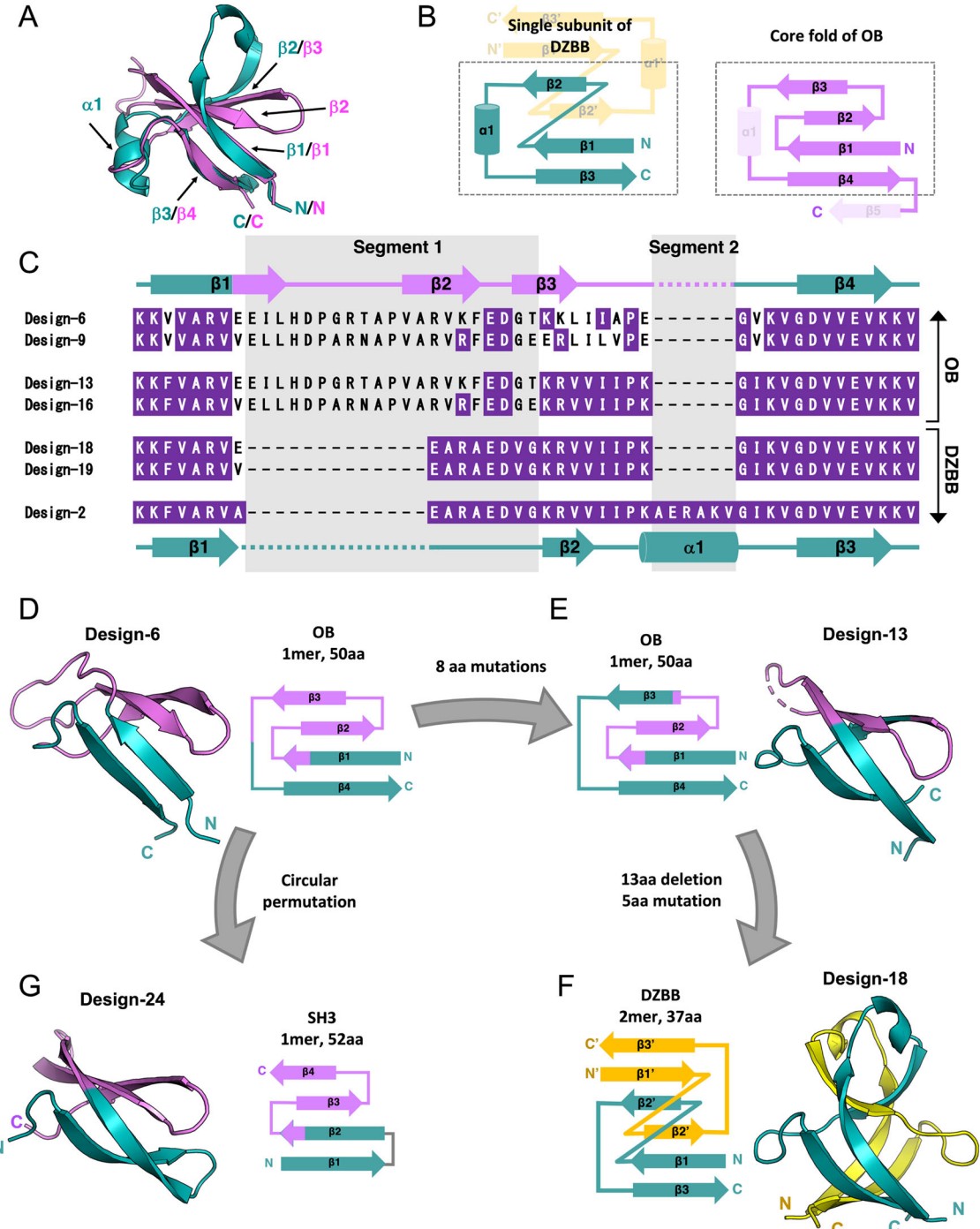

**Fig. 3 | Experimental reconstruction of the evolutionary pathway between the DZBB, OB, and SH3 folds. A** Superimposed structures of the single subunit of design-1 in the DZBB-fold and the OB domain of the ribosomal protein L2 from *Thermococcus kodakarensis*. **B** Comparison of the topologies of the DZBB and OB folds. In the OB fold, the non-essential secondary elements α1 and β5 are shown as dashed lines. **C** Multiple sequence alignment of the DZBB-OB chimeric proteins with the parent DZBB proteins, design-1 and design-2. The identical residues with design-2 are colored purple. The two significantly different segments between design-13 (OB) and design-2 (DZBB) are highlighted in gray. **D**–**G** Crystal structures and topological images of (**D**) design-6, (**E**) design-13, (**F**) design-18, and (**G**) design-24. The flows of the engineering procedures are shown as arrows.

design-24 adopted the four-stranded SH3 fold (Fig. 3G). This experimental conversion from OB to SH3 strongly supports the previous hypothesis that these folds could have emerged by a simple permutation of their four-stranded core fold[10].

## DNA binding abilities of the reconstructed β-barrels
As most β-barrels in the central dogma machinery function by interacting with nucleic acids, we investigated the DNA or RNA binding

capabilities of the reconstructed β-barrels by an electrophoresis mobility shift assay (EMSA) (Fig. 4 and Supplementary Fig. 20). When mixing the stable DPBB protein (design-0) and the 20 bp double-stranded DNA (dsDNA), some portion of dsDNA was slowed and stacked in the well as previously reported[7]. Most DNA molecules did not migrate from the well when mixed with the proteins with the stable DZBB fold (design-2, -18, and -19) and RIFT fold (design-3 and -4), indicating that these proteins formed large aggregates with dsDNA

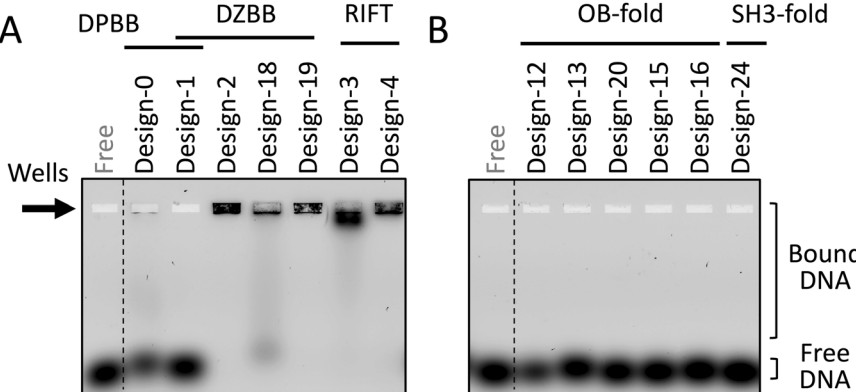

**Fig. 4 | Electrophoresis mobility shift assay to analyze the dsDNA binding properties of the reconstructed β-barrel proteins. A** 10 nM FAM-labeled dsDNA was mixed with 5 μM proteins with the DPBB, DZBB, and RIFT folds, and then the mixtures were subjected to 2% agarose gel electrophoresis. When design-2, -18, -19, -3, or -4 were mixed, most dsDNA molecules stacked on the wells. **B** The EMSA results for the proteins with the OB and SH3 folds. Two experiments were repeated independently with similar results. Source data are provided as a Source data file.

(Fig. 4A). In particular, design-2, -18, -19, -3, and -4 interacted with dsDNA even in high salt conditions (500 mM NaCl) (Supplementary Fig. 20B). We also performed the EMSA experiment with varied concentrations of design-2 and design-3 (Supplementary Fig. 20C and D). The mobilities of the dsDNA fragments were gradually slowed down as the protein concentration increased. Design-2 seems to have a higher affinity for dsDNA than design-3, as dsDNA shifted even at lower concentrations of design-2. Furthermore, no significant sequence specificity was observed when we tested another 20 bp dsDNA (Supplementary Fig. 20E). Design-2, -18, -19, -3, and -4 might interact with the phosphate groups in the DNA backbone in a similar way to the sulfate ions in the crystal structures of DZBB (Fig. 2B). The proteins with the DPBB, DZBB, and RIFT folds also interacted with ssDNA and ssRNA (Supplementary Fig. 20F and G). These findings suggest that, like their modern descendants, the ancient DPBB, DZBB, and RIFT proteins also interacted with nucleic acid polymers.

In contrast, the reconstructed OB and SH3 proteins did not interact significantly with any oligonucleotides (Fig. 4B and Supplementary Fig. 20H–J). Only high concentrations of design-13 and -16 retarded the migration of dsDNA slightly (Supplementary Fig. 20K). The acquisition of the weak DNA binding affinity of these two proteins may have resulted from additional lysine residues at the end of β3 and the 3rd loop, compared to other OB-fold chimeras (Fig. 3C). While the typical modern OB-fold proteins interact with an oligonucleotide at the surface of β1–3 (β2–4 in the SH3-fold proteins)[33], the corresponding region of ribosomal protein L2 used in this report (β2–3) does not directly interact with rRNA in the ribosome, likely resulting in the weak affinity of the reconstructed OB and SH3 proteins.

## Discussion

The fold conversion over evolutionary history is usually difficult to verify due to large diversities in sequences and structures between proteins with different folds. Still, some fold transition events have been detected by meticulous statistical methods[34,35]. Recent protein engineering endeavors also demonstrated that the relatively small number of point mutations foster the fold conversion, implying close relationships between distinct folds[36,37]. However, it might sometimes be misleading to decipher ancient protein evolution by referring only to the protein folds found today. In our study, a missing-link protein fold, DZBB, which is not found in modern proteins, offered a simple explanation for the evolutionary relationship between diverse β-barrel folds.

We demonstrated that the short and simple peptide, design-1, adopts to not only the homo-dimeric DPBB fold but also another homo-dimeric β-barrel fold, DZBB, like a metamorphic protein (Figs. 1, 2). It should be emphasized that the DZBB structure could not be predicted by AlphaFold2. It is still challenging for the state-of-the-art AI to predict metamorphic structures and protein folds outside of the program training set[38,39]. The comparison of the predicted model and crystal structures is discussed in detail in Supplementary note 1.

From the DZBB fold, the evolutionary pathway among distinct β-barrel folds, RIFT and OB, could be reconstructed by simple and feasible mutation steps. A single deletion in the DZBB sequence converted it into the RIFT fold (Fig. 1). In contrast, the single insertion of a short sequence forming a β-strand and a few point mutations converted it to an OB fold, accompanied by the oligomeric state change from dimer to monomer (Fig. 3). Furthermore, the reconstructed OB fold could also be converted to the four-stranded SH3 fold by a simple circular permutation (Fig. 3G). Thus, these ancient β-barrel folds (DPBB, RIFT, OB, and SH3) could be readily interconverted by a few feasible mutations through the missing-link fold, DZBB (Fig. 5), indicating their significantly close evolutionary relationships (Supplementary note 5).

This also implies that the diverse β-barrel folds can be produced from a limited sequence space. In an evolutionary time scale, the variety of ancient β-barrel folds might have diverged in a very short period, like the rapid diversification of animal species in the Cambrian. This rapid diversification of the various β-barrel folds probably preceded and primed the subsequent development of the elaborate molecular machines underlying the central dogma[40–44] (Fig. 5). Moreover, the reconstructed β-barrels (except for the mutant with the SH3 fold) retained the DNA and RNA binding affinities (Fig. 4 and Supplementary Fig. 20). Thus, during the diversification process of these folds, the fundamental nucleic-acid-binding property might have been inherited by the daughter folds, which then became specialized to the specific substrates and enzymatic reactions by stepwise mutations in each lineage.

Then, why was the DZBB fold lost in modern life? Interestingly, all modern DPBB and RIFT barrels exist as single-chain, pseudosymmetric proteins (except for the related eight-stranded barrel[6]). Their ancient genes would have duplicated and fused tandemly to form monomers. The structures of homo-dimeric DPBB and RIFT barrels show that the N-terminal end of one monomer is close to the C-terminal end of the other chain (Fig. 1B, E). This spatial arrangement could have readily allowed the polypeptide to fuse to the other chain without any dynamic fold rearrangement. Being single polypeptides, DPBB and RIFT could also have acquired more elaborate functions, performed by asymmetrically optimized residues and additional domains specifically linked to their N- and C-termini. In contrast, the N-terminal end of one chain and the C-terminal end of the other in the homo-dimeric DZBB are distant (Fig. 1C, D), making it impossible to integrate two short polypeptides into a monomer through a simple

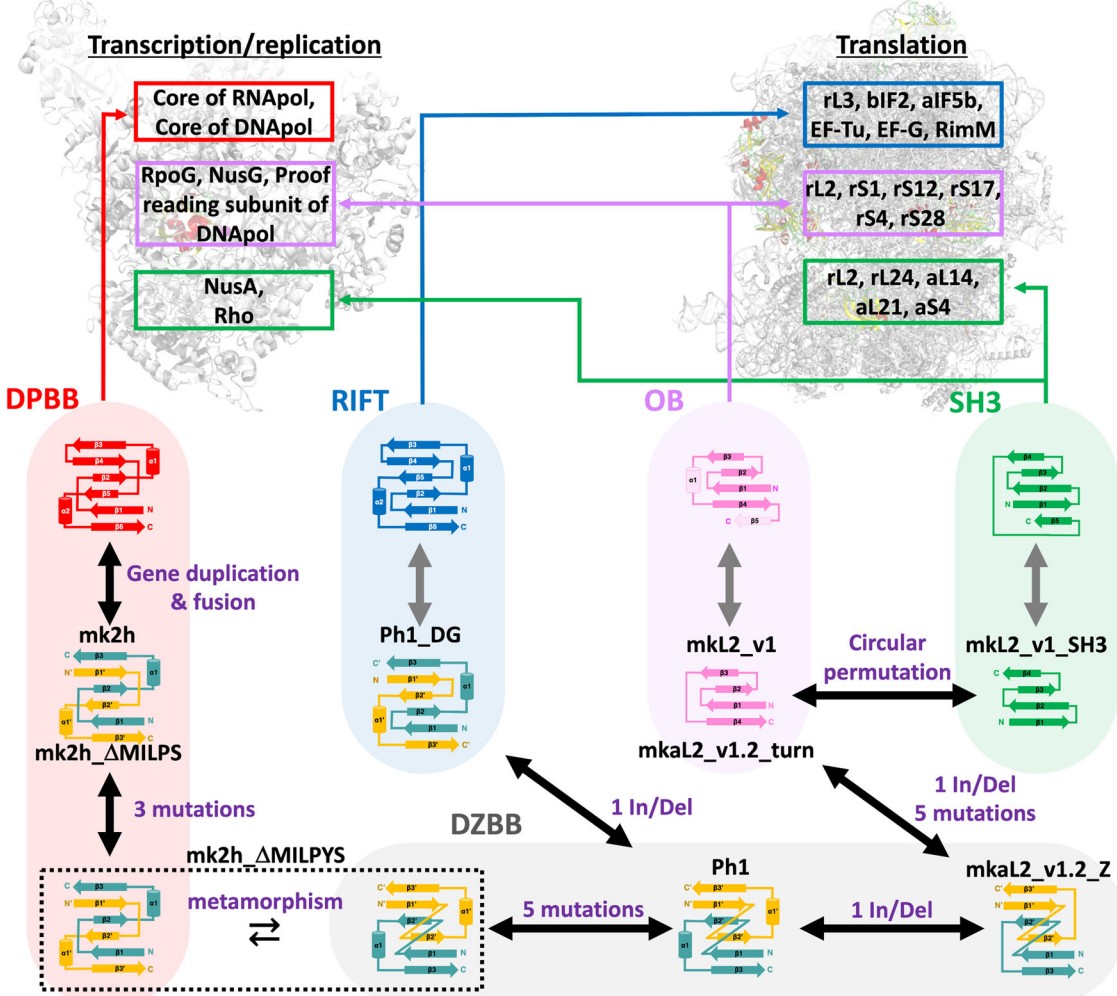

**Fig. 5 | The proposed evolutionary network between the modern four β-barrels through the missing link fold, DZBB.** The experimentally verified routes between each mutant pair are connected with black arrows, and the minimal necessary mutations between designs are written in purple. The DPBB protein's evolutionary pathway, which was examined in the previous study, is highlighted in a light red background. The evolutionary routes of DZBB, RIFT, OB, and SH3 are highlighted in gray, light blue, pink, and green backgrounds, respectively. Representative proteins conserving each barrel fold are shown in the upper panel.

gene fusion event. Because of this limited evolvability, the DZBB fold may have been outcompeted during the early evolution of life. The evolutionary pathways depicted here with the lost fold provide the groundwork for more detailed and broader studies of early protein evolution and the origin of the central dogma.

## Methods

### Construction of expression vector
The synthetic DNAs encoding the protein mutants used in this study, except for the design-3, -4, and -5 (Ph1_DG, Ph1_GG, and Ph1_GD), were purchased from Thermo Fisher Scientific and Integrated DNA Technologies and were amplified by PCR with the cloning_upstream and cloning_downstream primers (Eurofins Genomics; Supplementary Table 2). The genes encoding the design-3, -4, and -5 were constructed by the Splicing by Overlap Extension method with the mutant primers (Eurofins Genomics; Supplementary Table 2)[45]. The linear pET47b DNA was also amplified with pET47b_up and pET47b_down primers (Eurofins Genomics; Supplementary Table 2). Each PCR product was cloned into the pET47b vector for fusion with an N-terminal His-tag, by using an In-Fusion HD cloning kit (Clontech). The *Escherichia coli* DH5α strain was then transformed with the produced vectors. The resultant transformants were cultured on LB plates supplemented with 20 μg/ mL kanamycin (37 °C, overnight), and then used to inoculate to LB liquid medium and grown at 37 °C overnight. Each plasmid was extracted from the cells using a QIAprep Spin Miniprep Kit (QIAGEN). The sequences of the inserted genes were confirmed by Sanger sequencing.

### Protein expression and purification
*E. coli* BL21 Gold (DE3) cells (Agilent Technologies, CA) were transformed with the vectors harboring the genes of the respective protein mutants. The resulting transformants were cultured in 20 mL of LB medium supplemented with 20 μg/mL kanamycin (37 °C, overnight) and then inoculated into 2 L of LB medium (20 μg/mL kanamycin). After culturing the cells at 37 °C for two hours, 0.5 mM isopropyl b-D-1-thiogalactopyranoside (IPTG) was added to induce expression of the desired protein. The culture was continued under the same conditions for 4 hours. The cells were then harvested and stored at -20 °C.

The bacteria were disrupted by sonication in 60 mL of 50 mM potassium phosphate buffer, pH 6.5, and 150 mM NaCl. The lysate was centrifuged (4 °C, 11,000 x g, 20 min). The supernatant was filtered (0.45 μm pore-size) and then purified by HisTrap HP nickel affinity chromatography (GE Healthcare, IL). The N-terminal His₆-tags were cleaved with HRV-3c protease (Funakoshi, Japan) at 4 °C for 1–2 days. The treated samples were again loaded onto the HisTrap column, and

the flow-through fraction was recovered. The protein solutions were then loaded onto a HiLoad 16/600 Superdex 75 (GE Healthcare, IL) size exclusion chromatography column, equilibrated with 50 mM potassium phosphate buffer, pH 6.0, 150 mM NaCl. The purity of each protein sample was verified by SDS-PAGE. Because all of the designed proteins in this study lack tyrosine and tryptophan residues, the protein concentrations were determined by a BCA assay (Thermo Fisher Scientific), in which design-0 (mk2h_ΔMILPS) containing three tyrosine residues was used as the standard protein. The concentration of design-0 was determined by its absorbance at 280 nm.

## Biophysical characterization

The protein samples were prepared in 50 mM potassium phosphate buffer, pH 6.0, and 150 mM NaCl. Each protein's circular dichroism (CD) spectra were recorded from 200 to 250 nm at 20 °C, using a 0.1 cm path-length cell and a JASCO J820 circular dichroism spectrometer (JASCO, Japan). The protein concentrations were adjusted to approximately 20 μM. However, because the CD spectra of some mutants showed high tension (HT) voltages at the shorter wavelengths, their protein concentrations were diluted to 13–5 μM to reduce the HT voltages. The proteins with β-structures exhibit a variety of spectra patterns due to their structural diversity (e.g., parallel, left-twisted anti-parallel, and right-twisted anti-parallel arrangements)[46]. Still, they are well distinguishable from α-proteins and random coils.

To measure each protein's thermal stability, the ellipticity changes at 208 or 222 nm were monitored as the temperature was increased from 20 to 90 °C at a rate of 1.0 °C/min. The CD spectra were obtained at 20, 30, 40, 50, 60, 70, 80, and 90 °C. After cooling from 90 °C, the spectra at 20 °C were recorded again to verify the refolding ability.

We performed size exclusion chromatography to examine the protein foldability. Each purified protein (100 μL, 20 μM) was loaded onto a Superdex 75 increase 10/300 (GE Healthcare, IL) size exclusion column, equilibrated with 50 mM potassium phosphate buffer, pH 6.0, 150 mM NaCl, and run on an AKTA FPLC (Amersham Biosciences) at a flow rate of 0.75 mL/min.

## ANS fluorescence measurement

Design-1 (mk2h_ΔMILPYS) was diluted to 1 μM in the solutions containing various concentrations of malonates or ammonium sulfates (50, 100, 500, 1000, 1500, and 2000 mM). The pH values of the samples, including malonate or ammonium sulfate were adjusted to 7.0 and 6.0, respectively. Solutions of 2,000 mM potassium/sodium phosphate (pH 6.0), citrate (pH 7.0), acetate (pH 7.0), formate (pH 7.0), or glycine (pH 7.0) were also examined. The fluorescence probe 8-anilino-1-naphthalenesulfonic acid (ANS) was added (50 μM) and then the solution was placed in the dark for 30 min at room temperature. Using FP-8500DS fluorescence spectrometry (JASCO, Japan), the fluorescence spectra ranging from 400 to 650 nm were recorded with excitation at 380 nm.

## Crystallography

Before crystallization screening, we dialyzed all purified protein solutions against 20 mM Bis-tris HCl, pH 6.0, and 150 mM NaCl. The samples were then concentrated to 3–82 mg/mL. To screen the crystallization conditions for each protein, 96-well sitting-drop vapor-diffusion plates, and Wizard 1&2, Wizard 3&4, (Molecular Dimensions, United Kingdom), and Index HT (Hampton Research, CA) crystallization solutions were used. For crystallization, 0.2 μL of each protein was mixed in a 1:1 ratio with reservoir solutions and incubated at 20 °C. The conditions in which each protein formed crystals are listed in Supplementary Table 4. The cryo-protectant solutions were prepared with the reservoir solutions supplemented with 10–20% glycerol (Supplementary Table 4).

The X-ray diffraction data were collected at the Photon Factory[47,48] (Tsukuba, Japan), SPring-8[49–52] (Harima, Japan), or Swiss Light Source (Villigen, Switzerland). The beamlines are listed in Supplementary Table 4. The XDS program was used for the initial processing of diffraction data[53]. All crystal structures were solved by the molecular replacement method and refined with the program PHENIX[54,55]. The initial structure models for each mutant were determined by the MR phasing method, using phenix.phaser-MR. The model structures were updated manually using Coot and iteratively refined with Phenix.refine[56]. Statistics for diffraction data collection and refinement are summarized in Supplementary Table 5. 2Fo−Fc electron density maps of each crystal structures are shown in Supplementary Fig. 21.

## Electrophoresis mobility shift assay (EMSA)

For the EMSA, 5 μM of protein was mixed with 10 nM of FAM-labeled oligonucleotides (20 mM Tris-HCl, pH 8.0, 50 mM NaCl). The sequences of the DNA and RNA are shown in Supplementary Fig. 20A. The protein and oligo-nucleotide mixtures were incubated in the dark at room temperature for 10 min. After adding the loading dye (30% glycerol and bromophenol blue), the samples were fractionated on a 2% agarose gel (0.5 x TBE buffer). After electrophoresis, the DNA or RNA bands in the gel were imaged with an Amersham Typhoon scanner (GE Healthcare).

## Reporting summary

Further information on research design is available in the Nature Portfolio Reporting Summary linked to this article.

## Data availability

The atomic coordinate files are available in PDB. The accession codes are: 8JVN and 8JVO (design-1), 8JVP (design-2), 8JVQ (design-3), 8JVR (design-4), 8JVT (design-6), 8JVS (design-9), 8JVU (design-13), 8JVV (design-15), 8JVW (design-18), 8JVX (design-19), 8JVY (design-20), 8JVZ (design-24). The datasets generated during and/or analyzed during the current study are available from the corresponding author on reasonable request. Source data are provided with this paper.

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

## Acknowledgements

This work is based on experiments performed at KEK (project number: 2020G056 and 2022G005), SPring-8, and SLS. The authors are grateful to the beamline staff scientists at KEK, SPring-8, and SLS. We thank Hideaki Niwa, Toshiaki Hosaka, and Kentaro Ihara for assistance with the X-ray diffraction experiments. We also thank Hongding Liu for assistance of the DNA cloning experiment. S.Y. and S.T. were supported by JSPS (18H01328,

20K15854, and 22H01346). S.T. was also supported by the Astrobiology Center Program of National Institutes of Natural Sciences (AB0503).

## Author contributions

S.Y. and S.T. conceived and designed the experiments. S.Y. performed all designs and experiments. S.Y. and S.T. performed the crystallographic analysis. All authors discussed the results and jointly wrote and commented on the manuscript.

## Competing interests

The authors declare no competing interests.
