## [Peer Review File · Nature Communications]

An ancestral fold reveals the evolutionary link between RNA polymerase and ribosomal proteinsREVIEWER COMMENTS

Reviewer #1 (Remarks to the Author):

Yagi and Tagami provide the definitive experimental demonstration that the small beta barrel folds are readily interconvertible and thus, likely very likely related. Specifically, transitions between the OB fold and the SH3 fold, the DZBB fold and the OB fold, and the DPBB and the DZBB fold were described – with the DZBB fold being a newly discovered ‘bridge’ fold. Moreover, the sequence changes needed to achieve these transitions were relatively conservative – and in one case, dual folding was observed – highlighting the structural plasticity and fundamental traversability of the small beta-barrel folds. In short, this work is a tour de force support and I enthusiastically recommend publication in Nature Communications.

I provide some minor comments for the authors to consider at their discretion.

:: While the space of possible trajectories through fold space are likely greater than the space of explored solutions (i.e., biological history), the conservative mutations needed to make a DZBB fold would seem to suggest easy discovery. Thus, while I agree that interpreting the DZBB fold as a missing link is reasonable, I am left wondering why no DZBB folds seem to have persisted into the contemporary protein universe. Some comment on this point may be interesting.

:: The construct naming in this work is painful, even for a reader that has follows your work closely. Related to this point, it was at times confusing to sort out which constructs were folded in solution, versus needed high concentrations of ligand to fold, versus observed only in crystal structures.

Typos, etc.

: Line 53-54: <https://pubmed.ncbi.nlm.nih.gov/33502503/> may be a useful reference here

: Line 81: I think there is a typo in the name of your construct

: Lines 161, 162, 172, 209, 230, 270: b should be the Greek letter beta

Reviewer #2 (Remarks to the Author):

The two six-stranded β -barrel topologies DPBB and RIFT as well as the two five-stranded β -barrel topologies OB and SH3 constitute distinct folds but are nevertheless so similar that an evolutionary relationship between all four topologies has been proposed. This hypothesis is particularly attractive, because these β -barrels are found in DNA- and RNA-polymerases (DPBB and RIFT) and in ribosomal proteins and translation factors (OB and SH3) and therefore might have been essential for the realization of the dogma of molecular biology (DNA to RNA to protein) already early in evolution.

The goal of this work was to reconstruct the evolutionary pathways linking DPBB/RIFT/OB/SH3 through the interconversion of the different folds by means of protein engineering. The starting point for this endeavor was a simplified DPBB protein (termed mk2h_ΔMILPYS) that was constructed and characterized in a previous work of the same group (reference 1). The authors now unexpectedly found that, under different crystallization conditions, mk2h_ΔMILPYS did no longer fold as DPBB but in a novel topology termed DZBB, which classifies this protein as “metamorphic”. The inspection of the different crystal structures made the authors conclude that the binding of malonate induces the DPBB fold of mk2h_ΔMILPYS, whereas the binding of sulfate ions induces its DZBB fold. Interestingly, DZBB resembles the RIFT topology by sharing an all-anti-parallel strand pattern; however, the loop connecting β -strand 1 and β -strand 2 is rolled up in DZBB whereas it constitutes a simple β -turn in RIFT. Based on this observation, Ph1, a stabilized derivative of mk2h_ΔMILPYS with 5 mutations, could be converted into the RIFT fold through shortening the β 1- β 2 loop. Subsequently, the authors found that the crystal structure (and in part also the sequence) of the single subunit of homo-dimeric DZBB resembles the four-stranded core fold of the OB monomer but lacked an element corresponding to β -strand 2 of OB. The partial or full insertion of the corresponding sequence-stretch of OB into DZBB yielded chimeras (1st generation: tkol2_v1; 2nd generation: tkol2_v1.2) that adopted the OB core fold. The reverse engineering (insertion of a DZBB-specific sequence-stretch into OB) yielded a protein (tkol2_v1.2_Z) with the DZBB fold. Finally, based on the similar four-stranded core topologies of OB and SH3, circular permutation was applied to convert tkol2_v1 into mkaL2_v1_SH3, which adopts the SH3 fold. The different constructs were then tested by means of electrophoretic mobility shift

assays (EMSAs) for their ability to bind double-stranded and single-stranded DNA, as well as single-stranded RNA. The results indicate that DPBB, DZBB, and RIFT can interact with nucleic acids whereas OB and SH3 are unable to do so.

The results of this paper are indeed interesting, because the simple conversion of the metamorphic DPBB/DZBB topology into the RIFT and OB topologies by single deletions and insertions (and the straightforward conversion of OB into SH3 by a circular permutation) strongly support the evolutionary relationship of these different β -barrel folds. Moreover, the results suggest that these folds could have diverged from each other within a short period of time, which might have enabled the rapid emergence of sophisticated molecular machines involved in replication, transcription, and translation.

Despite this overall positive assessment of the presented work, there are a number of inconsistencies that have to be cleared up before publication.

Major points:

1. In contrast to what is stated in the text, the majority of the shown far-UV CD spectra do not indicate that the various generated protein variants are properly folded in solution:

Page 4, line 96f.: "...Ph1 folded with β -rich characteristics...(Fig. S4)". However, the far-UV CD-spectra of Ph1 (Fig. S4B, D, E) look weird and do not indicate the formation of β -strands (or other secondary structure elements), either in the presence or the absence of 500 mM ammonium sulfate.

Page 5, line 114f.: "CD experiments demonstrated that Ph1_GD and Ph1_DD folded (Fig. S7)...". However, the far-UV CD spectra suggest that these proteins fold only in the presence of 500 mM ammonium sulfate.

Page 6, line 151f.: "These experiments demonstrated that the small ions mediate the folding of mk2h_ Δ MILPYS,...". However, the far-UV CD-spectra shown in Fig. S8 look weird and do not indicate the formation of β -strands (or other secondary structure elements), independent of the presence of salts.

Page 7, line 191 f., page 8, line 200: with reference to Fig. S11 and Fig. S13, respectively, it is

claimed that the generated DZBB-OB chimeras adopt stable folds. However, the far-UV spectra shown in the two figures again look weird.

The strange shapes of the CD-spectra are somewhat surprising given that many of the investigated variants could be crystallized. My suspicion would be that the spectra are artifacts due to technical problems caused by a very high absorbance of the solutions. This high absorbance could be caused by salts, buffer components, or protein aggregation, to name just a few. In accordance with this assumption, the authors report about high tension voltages at shorter wavelengths (Supporting Information, page 3, line 73f.). I suggest that the authors use another method to demonstrate proper folding of their constructs in solution, preferably NMR.

2. The EMSAs shown in Fig. 4 are not very meaningful and just indicate that the DZBB and RIFT constructs somehow stick to double-stranded DNA. It even cannot not be excluded that the DNA “cross-links” the proteins, causing aggregation. To obtain meaningful information about the characteristics and the strength of the DNA-protein interaction, the authors should titrate the DNA with different protein concentrations, as is usually done with EMSAs. In addition, or alternatively, the authors could use other biophysical methods such as fluorescence polarization or NMR to obtain this information.

Minor points:

1. Figs. S4, S11, S13: SEC elution profiles: please add to the legend the association state of the proteins as deduced from the elution volume: is it always dimeric for DBPP and RIFT, and monomeric for OB and SH3?

2. The x-axis of the thermal unfolding curves in Fig. S11 and Fig. S13 are incorrectly labelled: “wavelength” needs to be replaced by “temperature”.

3. Page 7, line 179/Fig. 3C: it is not clear whether the insertions stemming from the OB fold were incorporated into mk2h_ΔMILPYS (as stated in the text) or into Ph1 (as suggested by Fig. 3C).

Reviewer #3 (Remarks to the Author):

In this beautiful work, Yagi and Tagami show that a reconstructed ancestor of small β -barrel folds displays metamorphic behavior. Not only do they solve both of its structures in different crystallization conditions, but they also convincingly pinpoint molecular triggers that foster the switch from one fold to another—no small task! Publication in Nature Communications is recommended after several minor revisions that contextualize their work more fully.

1) The authors should define what metamorphic proteins are (Murzin, Science 2008 and Kim and Porter, Structure 2021).

2) The authors do a nice job contextualizing their work relative to other studies done on small β -barrel folds. However, discussing the work of other groups on fold evolution will give readers a broader perspective on the problem. Specifically, the authors should discuss how their results relate to the evolutionary pathway recently discovered by Chakravarty and colleagues (Nat. Comm. 2023) and Farías-Rico and colleagues (Nat. Chem Biol. 2014). The recent engineering work of Bryan and Orban should also be discussed (Solomon, et al. PNAS 2023 and Ruan, et al. Nat. Comm. 2023).

3) Both forms of their metamorphic protein are domain swapped. Thus, a little discussion of domain swapping and how it might foster the metamorphism would also benefit readers: Koharudin and colleagues (PNAS 2013) and Bennett, Schlunegger, and Eisenberg (Protein Science 1995). The review by Dishman and Volkman (ACS Chemical Biology 2018) may also lend perspective.

4) If I understand correctly, AlphaFold2 predicted PH1's fold incorrectly. Discussion of this will benefit the field as there is a lot of confusion about AF2's ability to predict metamorphic proteins. The author's results are consistent with recent observations that AlphaFold2 fails to predict protein fold switching (Chakravarty and Porter, Protein Science 2022) and a new preprint demonstrating that AF2 struggles to predict the folds of proteins outside of its training set, which would include PH1

(<https://www.biorxiv.org/content/10.1101/2023.12.12.571380v1>). Please discuss.

We would like to thank the reviewers for their careful and thorough reading of this manuscript and thoughtful comments and constructive suggestions, which have helped to improve the quality of this manuscript. The following are our responses to the specific comments.

The following are our replies to Reviewer 1.

Specific comment 1:

While the space of possible trajectories through fold space are likely greater than the space of explored solutions (i.e., biological history), the conservative mutations needed to make a DZBB fold would seem to suggest easy discovery. Thus, while I agree that interpreting the DZBB fold as a missing link is reasonable, I am left wondering why no DZBB folds seem to have persisted into the contemporary protein universe. Some comment on this point may be interesting.

Reply: The reason why the DZBB fold does not persist into modern proteins was discussed only in the supplemental text in the previous manuscript. We have moved the relevant discussion to the main text (p.12, lines 312–325).

Specific comment 2:

The construct naming in this work is painful, even for a reader that has follows your work closely. Related to this point, it was at times confusing to sort out which constructs were folded in solution, versus needed high concentrations of ligand to fold, versus observed only in crystal structures.

Reply: According to the referee's advisement, we have simplified the construct

names (design-1 to design-24) in the manuscript and added the table summarizing their names and properties (Table S3).

Minor comment 1:

Line 53-54: <https://pubmed.ncbi.nlm.nih.gov/33502503/> may be a useful reference here

Reply: The reference is added on line 54.

Minor comment 2:

Line 81: I think there is a typo in the name of your construct

Reply: As we have renamed all constructs in the manuscript, the indicated typo was also removed.

Minor comment 3:

Lines 161, 162, 172, 209, 230, 270: b should be the Greek letter beta

Reply: We have corrected the style mistakes throughout the manuscript.

The following are our replies to Reviewer 2.

Specific comment 1:

In contrast to what is stated in the text, the majority of the shown far-UV CD spectra do not indicate that the various generated protein variants are properly folded in solution:

Page 4, line 96f.: "...Ph1 folded with β -rich characteristics...(Fig. S4)". However, the far-UV CD-spectra of Ph1 (Fig. S4B, D, E) look weird and do not indicate the formation of β -strands (or other secondary structure elements), either in the presence or the absence of 500 mM ammonium sulfate. Page 5, line 114f.: "CD experiments demonstrated that Ph1_GD and Ph1_DD folded (Fig. S7)..." . However, the far-UV CD spectra suggest that these proteins fold only in the presence of 500 mM ammonium sulfate.

Page 6, line 151f.: "These experiments demonstrated that the small ions mediate the folding of mk2h_ΔMILPYS,..." . However, the far-UV CD-spectra shown in Fig. S8 look weird and do not indicate the formation of β -strands (or other secondary structure elements), independent of the presence of salts.

Page 7, line 191 f., page 8, line 200: with reference to Fig. S11 and Fig. S13, respectively, it is claimed that the generated DZBB-OB chimeras adopt stable folds. However, the far-UV spectra shown in the two figures again look weird.

The strange shapes of the CD-spectra are somewhat surprising given that many of the investigated variants could be crystallized. My suspicion would be that the spectra are artifacts due to technical problems caused by a very high absorbance of the solutions. This high absorbance could be caused by salts, buffer components, or protein aggregation, to name just a few. In accordance with this assumption, the authors report about high tension voltages at shorter wavelengths (Supporting Information, page 3, line 73f.). I suggest that the authors use another method to demonstrate proper folding of their constructs in solution, preferably NMR.

Reply: As the referee commented, CD spectra from our designs are different from the typical CD spectra of β -proteins in most textbooks or reviews. Actually, it is known that β -proteins show a variety of spectra patterns because of their structural diversity (for example, parallel, left-twisted anti-

parallel, and right-twisted anti-parallel arrangements)(Micsonai et al., PNAS 2015). The diverse spectrum patterns obtained in this study are probably due to the variety of β -sheet structures. In contrast, α -proteins and random coils exhibit very typical CD spectra. Thus, we can at least say our CD spectra indicate the proteins are folded (not random coil), which is enough for our context as the detailed structures are determined by crystallography. Accordingly, we have modified the sentences (lines 97–98, 114, 157, 199, 208, 218, and 238 in the main manuscript and lines 74–77 in Supplemental information).

In addition, the buffer used in all CD experiments was 50 mM sodium phosphate pH 6.0 and 150 mM NaCl, and we believe that this basic buffer did not influence their CD spectra.

Specific comment 2:

The EMSAs shown in Fig. 4 are not very meaningful and just indicate that the DZBB and RIFT constructs somehow stick to double-stranded DNA. It even cannot not be excluded that the DNA “cross-links” the proteins, causing aggregation. To obtain meaningful information about the characteristics and the strength of the DNA-protein interaction, the authors should titrate the DNA with different protein concentrations, as is usually done with EMSAs. In addition, or alternatively, the authors could use other biophysical methods such as fluorescence polarization or NMR to obtain this information.

Reply: As the referee suggested, we have conducted the EMSA experiment with different protein concentrations. These results demonstrated that the DNA and protein complexes got gradually larger as the protein concentration increased. This implies that the interaction relies on the non-covalent bonds, not “cross-links.” The relative explanations were added on p.10, lines 254–258. Additionally, the gel images were also added to Fig. S19C and D.

Minor comment 1:

Figs. S4, S11, S13: SEC elution profiles: please add to the legend the association state of the proteins as deduced from the elution volume: is it always dimeric for DBPP and RIFT, and monomeric for OB and SH3?

Reply: The apparent molecular masses and oligomeric state of the proteins are summarized in Table S3. As the SEC experiment is not an accurate method for determining an absolute molecular mass (Golovchenko et al., *Journal of Chromatography* 1992), the estimated oligomeric state of proteins may not be precise. Indeed, even the folded proteins are estimated to be slightly larger than expected; for example, the apparent molecular mass of design-9 (OB fold) is 1.7 times larger than its theoretical monomer size, though it forms a monomer in the crystal.

Minor comment 2:

The x-axis of the thermal unfolding curves in Fig. S11 and Fig. S13 are incorrectly labelled: “wavelength” needs to be replaced by “temperature”.

Reply: According to the reviewer’s comment, we have corrected the label in the thermal unfolding curves in Fig. S11, 13, 14, 16, and 18.

Minor comment 3:

Page 7, line 179/Fig. 3C: it is not clear whether the insertions stemming from the OB fold were incorporated into mk2h_ΔMILPYS (as stated in the text) or into Ph1 (as suggested by Fig. 3C).

Reply: In this study, we have incorporated the insertions stemming from OB fold into mk2h_ΔMILPYS (design-1 in the revised manuscript), and the sequences were further engineered to resemble the more stable DZBB protein Ph1 (design-2). To clarify this procedure, we have modified the Fig. 3C, including the sequence of design-1.

The following are our replies to Reviewer 3.

Specific comment 1:

The authors should define what metamorphic proteins are (Murzin, Science 2008 and Kim and Porter, Structure 2021).

Reply: We have revised the manuscript by referring to the suggested papers in p. 5–6, lines 124–126.

Specific comment 2:

The authors do a nice job contextualizing their work relative to other studies done on small β -barrel folds. However, discussing the work of other groups on fold evolution will give readers a broader perspective on the problem. Specifically, the authors should discuss how their results relate to the evolutionary pathway recently discovered by Chakravarty and colleagues (Nat. Comm. 2023) and Farías-Rico and colleagues (Nat. Chem Biol. 2014). The recent engineering work of Bryan and Orban should also be discussed (Solomon, et al. PNAS 2023 and Ruan, et al. Nat. Comm. 2023).

Reply: We are grateful to the reviewer for the constructive suggestions. We have revised the manuscript to discuss the fold evolution with the suggested references (p. 10–11, lines 276–284).

Specific comment 3:

Both forms of their metamorphic protein are domain swapped. Thus, a little discussion of domain swapping and how it might foster the metamorphism would also benefit readers: Koharudin and colleagues (PNAS 2013) and Bennett, Schlunegger, and Eisenberg (Protein Science 1995). The review by Dishman and Volkman (ACS Chemical Biology 2018) may also lend perspective.

Reply: In accordance with the referee's suggestion, we have added a discussion regarding the domain-swapping and metamorphic properties of the proteins with the suggested references (p.6, lines 134–139).

Specific comment 4:

If I understand correctly, AlphaFold2 predicted PH1's fold incorrectly. Discussion of this will benefit the field as there is a lot of confusion about AF2's ability to predict metamorphic proteins. The author's results are consistent with recent observations that AlphaFold2 fails to predict protein fold switching (Chakravarty and Porter, Protein Science 2022) and a new preprint demonstrating that AF2 struggles to predict the folds of proteins outside of its training set, which would include PH1 (<https://www.biorxiv.org/content/10.1101/2023.12.12.571380v1>). Please discuss.

Reply: As the referee commented, the DZBB structure of mk2h_ΔMILPYS (denoted as design-1 in the revised manuscript) and Ph1 (design-2) could not be predicted correctly by AlphaFold2. Additionally, we have submitted their sequences as the targets of the 15th Critical Assessment of Techniques for Protein Structure Prediction (CASP15). As no team could reliably predict our structures as their first candidates, such drastic rearrangements induced by mutations were regarded as one of the most challenging and unsolved topics in protein structure prediction in CASP15. Therefore, these structures might serve as good training models for developing protein structure prediction programs. The relevant discussion was added to the p.11, lines 287–291.

REVIEWER COMMENTS

Reviewer #2 (Remarks to the Author):

Regarding my two original major points (named 'specific comments' by the authors), I am not yet fully convinced:

1. Based on their intensity minimum close to 200 nm, a number of the shown far-UV CD spectra clearly indicate a random coil structure (for example Fig. S11B,C,E,F; Fig. S13C, Fig. S18A). I would still recommend that another method is used to show that the constructs are folded in solution. Simple 1D-NMR would probably do.
2. While I am satisfied with the EMSA data shown in Fig. S19C,D, I am confused by Fig.4: no bands are visible in either of the slots; moreover, it seems as if the gel was upside down.

Reviewer #3 (Remarks to the Author):

The authors have done beautiful work and addressed all of my comments well. I highly recommend this manuscript for publication.

One minor comment: the authors might consider adding their experimentally determined structure of design-2 to Figure S3 so that readers may easily compare the predicted and experimentally determined structures.

We would like to thank the reviewers for their careful and thorough reading of this manuscript and thoughtful comments and constructive suggestions, which have helped to improve the quality of this manuscript. The following are our responses to the specific comments.

The following are our replies to Reviewer 2.

1. Based on their intensity minimum close to 200 nm, a number of the shown far-UV CD spectra clearly indicate a random coil structure (for example Fig. S11B,C,E,F; Fig. S13C, Fig. S18A). I would still recommend that another method is used to show that the constructs are folded in solution. Simple 1D-NMR would probably do.

Reply: Our manuscript contains not only successfully folded designs but also unsuccessful ones. The CD spectra in S11B, C, E, F; Fig. S13C, Fig. S18A are from the unsuccessful designs (design-7, 8, 10, 11, 14, 23). They were indicated to be random coils by CD as we mentioned in our manuscript. To avoid confusion for readers, we updated Table S3 to clarify which protein designs are folded or random coils.

2. While I am satisfied with the EMSA data shown in Fig. S19C,D, I am confused by Fig.4: no bands are visible in either of the slots; moreover, it seems as if the gel was upside down.

Reply: The direction of the gel image is correct. You can see the loading wells at the top of the gel image, which are now indicated by a black arrow. In the lanes for design_2,18,19,3, and 4, the DNA molecules likely formed large complexes with the mixed proteins, then they were stacked around the loading wells.

The following is our reply to Reviewer 3.

One minor comment: the authors might consider adding their experimentally determined structure of design-2 to Figure S3 so that readers may easily compare the predicted and experimentally determined structures.

Reply: We have added the experimentally determined structure of design-2 to Figure S3.

REVIEWERS' COMMENTS

Reviewer #2 (Remarks to the Author):

The authors have clarified my issues regarding the CD spectra. However, I still do not see any bands in Figure 4 except the one for free DNA.